# Evaluation of New Octacalcium Phosphate-Coated Xenograft in Rats Calvarial Defect Model on Bone Regeneration

**DOI:** 10.3390/ma13194391

**Published:** 2020-10-01

**Authors:** Yoona Jung, Won-Hyeon Kim, Sung-Ho Lee, Kyung Won Ju, Eun-Hee Jang, Sung-O Kim, Bongju Kim, Jong-Ho Lee

**Affiliations:** 1Clinical Translational Research Center for Dental Science, Seoul National University Dental Hospital, Seoul 03080, Korea; loveme2992@snu.ac.kr (Y.J.); kwh90@sju.ac.kr (W.-H.K.); shlee79@snu.ac.kr (S.-H.L.); jukyungwon@snu.ac.kr (K.W.J.); 2Graduate School Department of Dental Science, Seoul National University, Seoul 03080, Korea; 3Department of Mechanical Engineering, Sejong University, Seoul 05006, Korea; 4Department of Oral and Maxillofacial Surgery, School of Dentistry, Seoul National University, Seoul 03080, Korea; 5Chiyewon Co., Ltd., Gyeonggi-do 11927, Korea; jeh@chiyewon.com (E.-H.J.); ceo@chiyewon.com (S.-O.K.)

**Keywords:** xenograft, animal experiment, bone regeneration, rat calvarial defect, octacalcium phosphate coating

## Abstract

Bone graft material is essential for satisfactory and sufficient bone growth which leads to a successful implant procedure. It is classified into autogenous bone, allobone, xenobone and alloplastic materials. Among them, it has been reported that heterogeneous bone graft material has a porous microstructure that increases blood vessels and bone formation, and shows faster bone formation than other types of bone graft materials. We observed new bone tissue formation and bone remodeling using Ti-oss^®^ (Chiyewon Co., Ltd., Guri, Korea), a heterologous bone graft material. Using a Sprague–Dawley rat calvarial defect model to evaluate the bone healing effect of biomaterials, the efficacy of the newly developed xenograft Ti-oss^®^ and Bio-Oss^®^ (Geistilch Pharma AG, Wolhusen, Switzerland). The experimental animals were sacrificed at 8 and 12 weeks after surgery for each group and the experimental site was extracted. The average new bone area for the Ti-oss^®^ experimental group at 8 weeks was 17.6%. The remaining graft material was 22.7% for the experimental group. The average new bone area for the Ti-oss^®^ group was 24.3% at 12 weeks. The remaining graft material was 22.8% for the experimental group. It can be evaluated that the new bone-forming ability of Ti-oss^®^ with octacalcium phosphate (OCP) has the bone-forming ability corresponding to the conventional products.

## 1. Introduction

The bone graft material is essential for satisfactory and sufficient bone growth which leads to successful implant procedures [1]. This material should maintain space for new bone growth and bone fusion. An ideal bone graft material should ultimately develop new bones while simulating human bone remodeling methods, based on its excellent biocompatibility. It is classified into autogenous bone, allobone, xenobone and alloplastic materials. Among these, autologous bone graft could satisfy these conditions most closely. Despite its many advantages, its use is limited due to donor defects and complications, rather low volume stability problems, and restrictions in supply and demand which is fatal. Limitation in supply frequently occurs in allograft as well [2,3]. Therefore, to smoothly manage the supply and demand of bone graft materials, it is essential to develop a graft material technology using heterogeneous bones.

Xenograft materials are produced by removing organic matter from bones of cow, horse and pig through deproteinization. Its advantage is that it has a similar porous structure as human cancellous bone, is high in osteoconduction as its slow absorption helps to maintain space, and is competitive in price compared to other bone graft materials. In terms of bone healing mechanism, the bone formation ability of xenograft is generally lower than that of autologous or allogenic bones, but some research reported that the porous microstructure increases blood vessels and bone growth [4] and xenograft shows more rapid bone formation than other types of bone graft [5]. Among these xenograft materials, a deproteinized bovine bone materials (DBBM) is the most widely used clinical product for its stabilized and excellent bone formation ability [6].

However, most previous studies reported that DBBM has excellent volume retention and bone conductivity but has a very low absorption rate in vivo [7,8], and the amount of new bone formation is lower than that through natural healing [9]. The volume may be maintained with the low absorption rate of DBBM, but too little absorption can delay new bone formation [7,8,9].

In order to improve these problems, development of materials with high biocompatibility that can reflect the characteristics of human bone remodeling by combining biodegradable properties is required, such as bioactive ceramics (hydroxyapatite (HA), β-tri-calcium phosphate (β–TCP), OCP, polyesters (polylactic acid (PLA), polycaprolactone (PCL)) or polyhydroxyalkanoates-based biodegradable polymers extracted from microorganisms [10].

Among biodegradable materials, OCP has come to attention as a material that can complement the characteristics of conventional bone graft materials. OCP consists of a precursor of biological apatite crystals similar to bones and teeth [11,12], and it has a physical property that is converted into HA by dissolving in ionic and pH concentration similar to that of human plasma [13]. These characteristics promote osteoblast differentiation and can induce new bone formation [14]. Previous studies reported that OCP coating and OCP-based materials selectively adsorb serum components to OCP at the beginning of transplantation in the rat calvarial defect model [15], and due to its high bone conductivity, its’ bone formation ability is close to that of autogenous bone [16].

In order to apply these advantages of OCP to medical devices, there were studies that evaluated the effect of OCP coating on titanium alloy (Ti6Al4V) implants [17] or bone regeneration on synthetic OCP, but most of them were basic experimental studies on OCP materials [18], and research compared with dental bone graft materials is insufficient.

Therefore, this study applied Ti-oss^®^ which is DBBM surface-treated with newly developed OCP, and Bio-Oss^®^ which is the most widely used product in the clinical field, on a rat calvarial defect model to analyze the bone regeneration properties of OCP through observing its biocompatibility, new bone tissue-forming ability and degree of bone remodeling through grafting site exposure, inflammation and complications.

## 2. Materials and Methods

### 2.1. Preparation of Animals

In our animal experiment, twenty-four 6 week-old female Sprague Dawley rats (150–200 g, Orient Bio, Gapyeong, Korea) were used, and six rats were applied per group. The rats were bred in Specific-Pathogen-Free (SPF) laboratory which maintains constant temperature (21 °C ± 1 °C), humidity (55%) and 12 hours light:dark cycle (light: 07:30–20:00, dark: 20:00–07:30), and they were supplied with normal feed (Purina Rodent Chow, Purina Co., Seoul, Korea) and water. Our study was approved by the Institutional Animal Care and Use Committee of Seoul National University (IACUC; approval no. SNU-180727-7-4), conducted in accordance with the ARRIVE guidelines from 2018 to 2019.

### 2.2. Specimen Preparation

Using a rat calvarial defect model to evaluate the bone healing effect of biomaterials, the efficacy of the newly developed xenograft Ti-oss^®^ was evaluated (Figure 1a). The Ti-oss^®^ used was the same type as the specimen applied in the previous study [14]. The Ti-oss^®^ was produced by coating OCP on the surface of DBBM material extracted from cow bone through LeGeros method [19]. To remove the fat and proteins from the OCP-coated DBBM material, meticulous cleansing and heat treatment techniques were applied. Through this manufacturing process in the previous study, the calcium core structure of the cancellous bone was reduced to about 15% which increases the pore size [14]. The Bio-Oss^®^ control group and the Ti-oss^®^ experimental group were compared and analyzed (Figure 1a,b). A total of 24 animals were used in the experiment which were randomly assigned as 12 per group. 6 animals were sacrificed at 8 and 12 weeks. During the recovery period, any exposure to the transplant site, inflammation, or other complications were observed. The range of pore size of Ti-oss^®^ was 0.5–1 mm and Bio-Oss^®^ was 0.25–1 mm.

### 2.3. Surgical Procedures

After 1 week of the quarantine period, the rats were anesthetized by intraperitoneal injection using a 3 ml mixture (100 mg/kg) of pentobarbital (Hanlim Pharm, Co., LTD, Gyeonggi, Korea) and chloral hydrate (Sigma-Aldrich. Co., ON, Canada). The hair was removed from the head and it was fixed after disinfection using povidone-iodine(Green Pharmaceutical Co., Ltd., Seoul, Korea). The surgical site was anesthetized by infiltration with 2% lidocaine(Dai Han Pharm Co., Ltd., Seoul, Korea), and the incision was performed along the median approximately 1.0 cm to 1.5 cm from the anterior to the posterior part of the frontal bone. The frontal valve was lifted to expose the upper surface of the skull. On the top of the exposed skull, a circular defect with a diameter of 5 mm was formed using an 8 mm inner diameter trephine bur to produce a rat calvarial defect model (Figure 2a). After applying the control and experimental groups on the defect site (Figure 2b) [20], the surgical site was closed in layers and sutured using an absorbent suture (Vicryl 3-0, 4-0; Ethicon Inc., NJ, USA).

### 2.4. Specimen Preparation and Histomorphormetric Analysis

The experimental animals were sacrificed at 8 and 12 weeks after surgery for each group and the experimental site was extracted. The extracted site was fixed in 4% neutral buffer formalin(MERCK 104005, Merck KGaA, Darmstadt, Germany) for 10 days, demineralized with pH 7.4 ethylenediaminetetraacetic acid(EDTA) decalcifying solution(Milestone Medical, Sorisole BG, Italy) for 7 days and embedded in paraffine(Dynebio INC., Gyeonggi, Korea). The 8 µm thick specimen was sectioned at the middle part of the experimental site, and Hematoxyline-Eosin stainings were performed to observe at 100× and 200× with an optical microscope(Olympus Co., Seoul, Korea). The 100× magnified tissue specimen images were reproduced on the computer for histometric analysis of the new bone. The area of the defective new bone (mm^2^) was to the boundary of the newly formed bone, including mineralized and residual bone graft material, bone marrow and fibrous connective tissue, and neovascularization. The bone density (%) was the ratio of only the mineralized bone from the total area of the new bone, excluding other tissues. The histometric analysis was performed by a professionally trained blind investigator. To evaluate the formation of new bone, the cross-sectional area of new bone, residual graft material and connective tissue within 8 mm defect was measured, respectively. Then, the calculation formula from previous literatures (new bone area percent (%) = new bone area (mm^2^)/(new bone area (mm^2^) + residual graft materials (mm^2^) + connective tissue (mm^2^)) was used to perform comparative evaluation on bone formation (Figure 3) [21,22,23,24].

### 2.5. Statistical Analysis

The sample size was based on the statistical method presented by IACUC which was alpha error 0.05, beta error 0.2, 50% graft materials between each group and mean deviation [25]. The minimum quantity of 6 animals per group and a total of 24 animals were considered valid at a 5% significance level and 80% power. Each area of the new bone, residual graft and connective tissue were calculated using the formula from previous literature, and it was statistically analyzed using Sigmaplot 14 statistics program (Systat Software Inc., San Jose, CA, USA). One-way ANOVA was conducted followed by Fisher LSD, and the p-value was set to 0.05.

## 3. Results

### Tissue Morphology Analysis Results

The average new bone area for the Ti-oss^®^ experimental group at 8 weeks was 17.6% which was 0.3% higher than the Bio-Oss^®^ control group (17.3%), but there was no statistically significant difference. The remaining graft material was 26.3% for the control group and 22.7% for the experimental group (Figure 4).

The average new bone area for the Ti-oss^®^ group was 24.3% which was 0.7% lower than the Bio-Oss^®^ group (25.1%) at 12 weeks, but there was no statistically significant difference. The remaining graft material was 29.8% for the control group and 22.8% for the experimental group (Figure 4).

No abnormalities such as infection or complications after surgery were found in all groups. Histometric analysis, quantitative evaluation of bone formation and tissue type measurement analysis were conducted for the control and experimental group samples produced at 8 and 12 weeks sacrifice.

At 8 weeks, new bone formation was observed in bone defects applied with Bio-Oss^®^ and Ti-oss^®^ (Figure 5a,c). Newly formed bone is attached to the graft materials and show active remodeling with harversian structures (Figure 5b,d). New bone was observed to be formed adjacent to the graft materials (Figure 5b,d). No inflammatory tissue was observed in both groups, but it was observed that connective tissue was filled between the new bones (Figure 5a,c).

At 12 weeks, osteoblasts were observed in both two groups of bone defects applied with Bio-Oss^®^ and Ti-oss^®^, and the binding between the new bone and graft materials was observed to be closer compared to at 8 weeks (Figure 6). Newly formed bone is tightly attached to the graft materials and shows numerous reversal lines. Some areas show characteristics of less-matured bone areas, which show larger osteocyte lacuna. The bone marrow was filled with loose connective tissue. The new bone was observed to have completed bone remodeling and there is little inflammatory cell infiltration (Figure 6). The new bone density was observed to be high, and connective tissue was observed between the new bones, similar to at 8 weeks (Figure 6a,c).

## 4. Discussion

This study was designed to observe a rat calvarial critical defect model with xenograft for 8 and 12 weeks and evaluate the volume of newly generated bone tissue through tissue morphological analysis. A critical defect is defined as the smallest bone defect of a particular animal that does not heal spontaneously over its lifetime [26,27], and the size of the defect is important in evaluating biomaterials for bone regeneration [2,28]. The size of the critical defect of the rat calvarial model was 8mm in diameter from a previous study [3], so this study also designed a defect model of the same size.

Cow bone is used as graft material in the calvarial defect after complete removal of the organic material using ethylendiamine, leaving a porous crystalline hydroxyapatite structure (DBBM) [29]. Although there are concerns about bovine spongiform encephalopathy (BSE) among some countries and clinicians and does not have osteogenesis ability like autogenous bone graft, it is excellently evaluated for its bone conduction, ease of manipulation and its ability to act as a scaffold in the process of bone formation, allowing lacunenae to exist from an ultrastructure perspective. Most of all, it has trust in safety as it has been used clinically for a long time [30,31].

Bio-Oss^®^ is the most clinically used product. Its particle size is 250–1000 µm [32] and has a 75~80% porous structure [33]. It has been evaluated for its stable volume maintenance, excellent bone conduction and effective bone regeneration in animal and human bone regeneration studies [8].

In the case of xenograft materials such as Ti-oss^®^ and graft materials combined with calcium phosphate ceramic materials such as synthetic HA or β-tricalcium phosphate (β-TCP), the characteristics of calcium phosphate ceramic materials may affect the process of bone formation, so further study on the material is required.

The synthetic HA is classified as a relatively chemically-stable substance, and it does dissolve and remains in the bone defect area for a long time [34]. Some studies reported that HA ceramic particles can affect the biological response of fibroblasts and myoblasts [35].

In contrast, β –TCP causes an osteoclastic cellular phagocytotic response after chemical dissolution due to its properties of dissolving at physiological pH when grafted on the bone. Under these physiological conditions, biodegradation chemically or by osteoclasts occurs and new space is formed. This space is used as a scaffold for forming osteoblast colonies [36], which is ultimately beneficial for bone regeneration [34]. It has been observed that β –TCP causes a2 integrin subunit gene expression and activation of the mitogen-activated protein kinase (MAPK)/extracellular related kinase (ERK) signaling pathway [37].

The OCP coated on the surface of Ti-oss^®^ is assumed to be a precursor of biological apatite crystal of dentin, enamel and bone [11] and along with chemical degradation, biodegradation by osteoclast-like cells occurs in the physiological environment [38].

The OCP is gradually converted into the hydroxyapatite phase in many studies that have performed X-ray diffraction analysis on the OCP grafted on bone and subcutaneous tissue [39,40,41,42]. The OCP crystal consumes calcium ions and releases inorganic phosphate ions simultaneously in the physiological environment [43], and it is known that fluoride ions promote hydrolysis even in very small amounts in physiological fluids [44]. In a physiological environment, OCP and OCP hydrolysate facilitate the adsorption of circulating serum proteins such as a2HS-glycoproteins [42]. In particular, protein adsorption involved in bone metabolism, such as apoprotein, is known to promote bone regeneration [45,46].

In this study, no statistically significant difference was observed between the Ti-oss^®^ and Bio-Oss^®^ groups for both 8 and 12 weeks. For Ti-oss^®^, the biodegradability of the material was improved due to the OCP coating, so the amount of residual graft material tended to be less than that of Bio-oss^®^, but there was no statistically significant difference. Previous studies also reported that synthetic OCP and OCP-based material are highly biodegradable, and these reports showed a similar trend with our results [33,47,48]. A study by Kim et al., analyzed the products Bio-Oss^®^ and Bongros^®^ (Bio@ Inc., Seongnam, Korea) at 4 and 8 weeks after surgery, and it was observed that Bio-Oss^®^ (28.5%) showed less bone formation than Bongros^®^ (42.9%) at 4 weeks, but at 8 weeks, Bio-Oss^®^ (54.1%) showed more bone formation than Bongros^®^ (50.9%) [49]. The results of this study did not show any statistically significant difference between the 2 groups at 8 and 12 weeks, but unlike the results at 8 weeks, the bone formation ability of Bio-Oss^®^ showed a slightly increasing tendency compared to Tio-oss^®^ at 12 weeks. This showed a similar trend with the study results by Kim et al. [49].

In previous research considering the characteristics of bone formation by OCP, after biodegradation of OCP by osteoclast-like cells followed by OCP bone regeneration, new bone deposition in the structure composed of osteoblasts, OCP particles and non-collagenous proteins was observed in various animal model studies [50,51,52]. In the early stages of OCP-HA conversion, OCP-based material induces biological activation for bone regeneration through the physicochemical conversion process [40,41], and it suggested that the degree of bone resorption of OCP is related to the amount of new bone stimulated by OCP [53].

However, histometric analysis was performed in this study after 8 and 12 weeks of transplantation in a rat calvarial model to evaluate the bone-forming ability of xenograft materials. Analysis of the pore structure and porosity of the xenograft material and additional evaluation to compare the characteristics of the graft material after bone formation through micro-CT are considered necessary in further studies. It is considered that further studies should include bone defect models with various clinical conditions, securing space for tissue regeneration after bone graft and the effect of maintenance.

## 5. Conclusions

The bone-forming ability of Ti-oss^®^ on a rat calvarial model did not show any statistically significant difference compared to the Bio-Oss^®^ control group at both 8 and 12 weeks. Based on our results, it can be evaluated that the new bone-forming ability of Ti-oss^®^ with OCP has the bone-forming ability corresponding to the Bio-Oss^®^.

## Figures and Tables

**Figure 1 materials-13-04391-f001:**
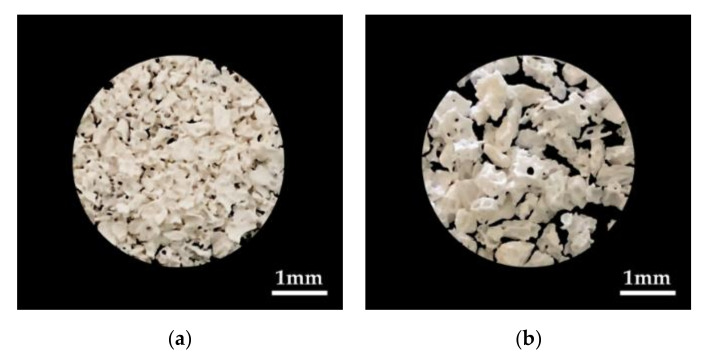
The bone graft materials used in our study. (**a**) Ti-oss^®^; (**b**) Bio-Oss^®^.

**Figure 2 materials-13-04391-f002:**
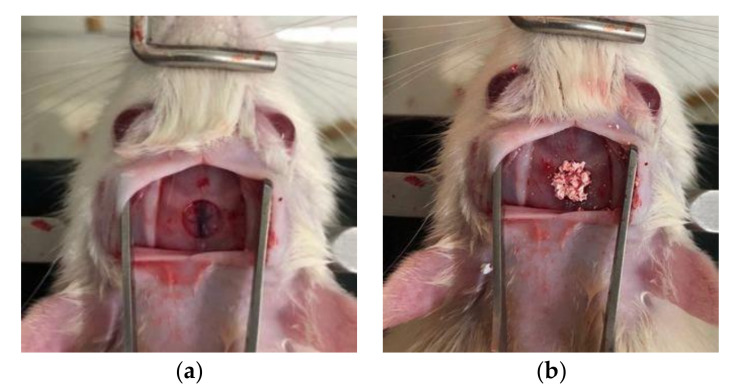
Calvarial defect in rats. (**a**) The appearance of calvarial defect used trephine bur, (**b**) Bone graft applied to calvarial defect.

**Figure 3 materials-13-04391-f003:**
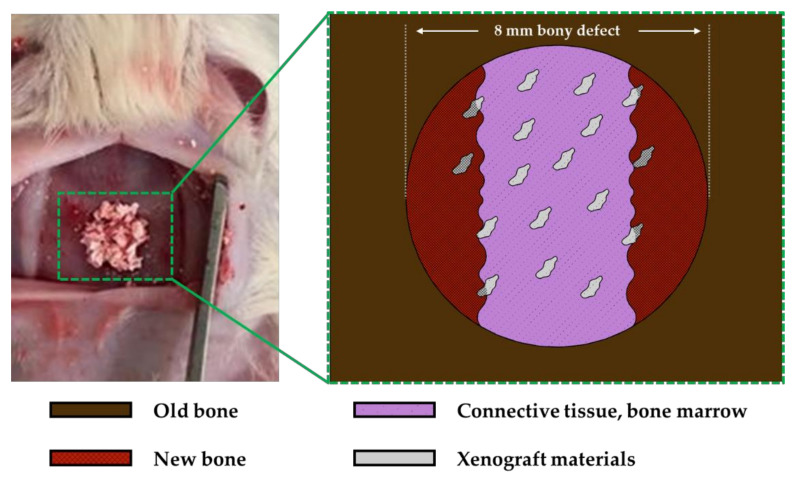
Schematic drawing of animal model for histometric analysis.

**Figure 4 materials-13-04391-f004:**
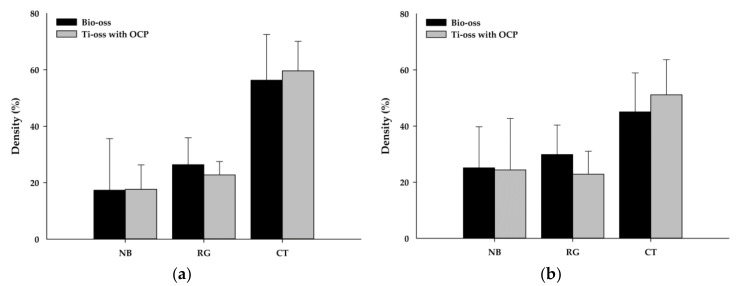
The percentages of the new bone density, residual graft materials and connective tissue for each group (*n* = 6 per group). (**a**) results at 8-weeks post-surgery; (**b**) results at 12-weeks post-surgery. Note: NB: New Bone; RG: Residual Graft Material; CT: Connective Tissue.

**Figure 5 materials-13-04391-f005:**
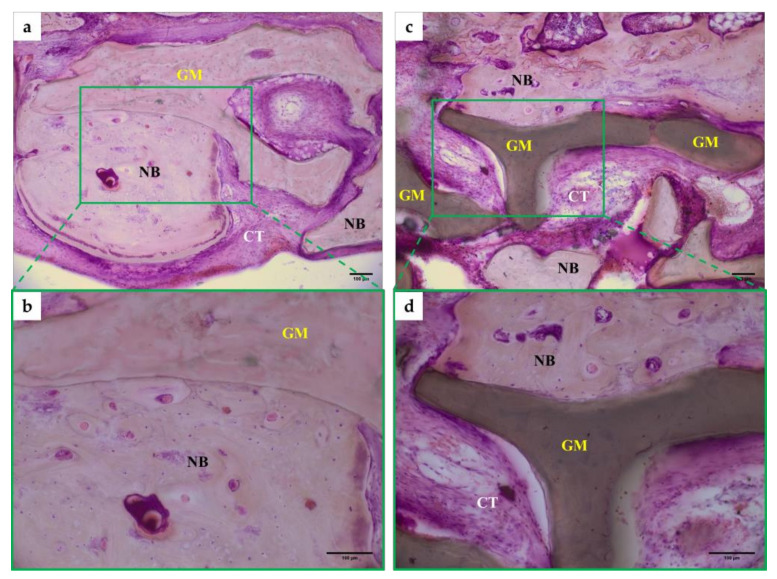
Image of Histomorphormetric analysis in a rat calvarial at 8 weeks. (**a**,**b**) Bio-Oss^®^ at 8weeks, (**c**,**d**) Ti-oss^®^ at 8 weeks. All groups was H&E staining and observed ×100 and ×200 with an optical microscope. NG: New Bone, GM; Graft Material, CT; Connective Tissue.

**Figure 6 materials-13-04391-f006:**
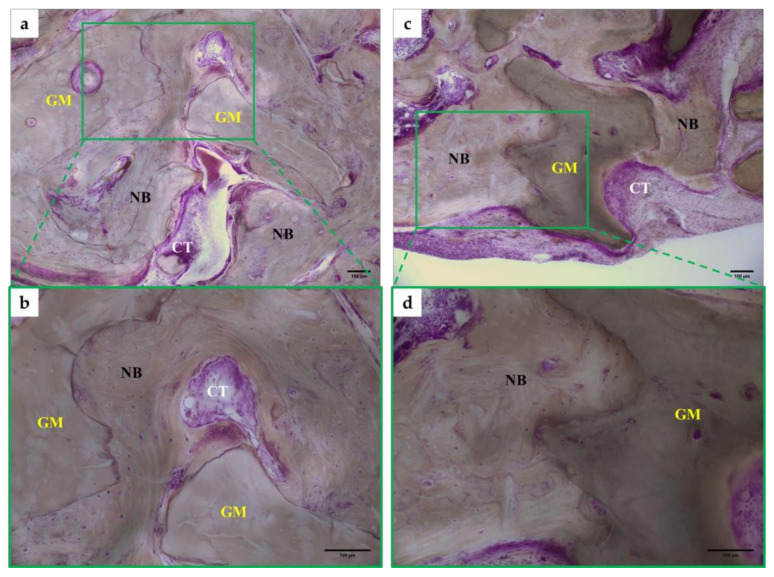
Image of Histomorphormetric analysis in a rat calvarial at 12 weeks. (**a**,**b**) Bio-Oss^®^ at 12 weeks, (**c**,**d**) Ti-oss^®^ at 12 weeks. All groups were H&E-stained and observed at ×100 and ×200 with an optical microscope. NG: New Bone, GM; Graft Material, CT; Connective Tissue.

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
