# Peer review of "Evaluation of New Octacalcium Phosphate-Coated Xenograft in Rats Calvarial Defect Model on Bone Regeneration"

_materials, 2020, doi:10.3390/ma13194391_

Round 1

Reviewer 1 Report

Dear Sirs,

Well written paper,however some few comments.

You claim to have no significant differences. Was this only due to the number of animal used? Six is quite little, thus did you do a power analysis in order to know how many is needed to draw such a conclusion? Please include.

Please also include microCT, both to quantify the pore structure of Ti-Oss and above the animal studies.

The introduction needs to rewritten, there is a lack of connections between the sentence structure and also more citations are needed.

Reviewer 2 Report

The paper by Yoona Jung et al. entitled ‘The Effect of Octacalcium Phosphate Coated 2 Xenograft in Rat Calvarial Defect Models on Bone 3 Regeneration’ describes comparison of two commercially available bone substitutes (Bio-Oss® and Ti-oss®) in healing critical defects in rat models. The paper is quite difficult to read, it is not clear what type of research is reported and what are the results. It seems that it is just a simple comparison of two existing commercial products with the result that there is no significant difference.

Major points

  1. The title is misleading. No effect of OCP coating is reported.
  2. Abstract is unclear.

„Using a Sprague-Dawley rats calvarial defect model to evaluate the bone healing effect of biomaterials, the efficacy of the newly developed xenograft Ti-oss® and Bio-oss®.“  

„The average new bone mass for the Ti-Oss® experimental group at 8 weeks was 17.6%.“

It is not stated how the new bone mass is quantified to be expressed as %.

  1. Most of the introduction is about auto-, allo-, xeno- materials. I do not see any reason for that as it is not related to the research reported. Some parts of the Discussion section would be more suitable for the introduction.
  2. The authors operate with the terms ‚new bone area‘, ‚new born area‘, ‚new bone mass‘ meaning probably the same.
  3. There is no comparison with other existing products. Discussion section should contain such comparison.

Minor points

  1. There are many undefined abbreviations throughout the manuscript (HA, PLA PCL)
  2. Some references are doubled. 31 = 40   32 = 42
  3. Carefully check the spelling. cadevaric   Histomorphormetric

     capital letters ?   defect in Rats   appearance of Calvarial defect

  1. „Sprague-Dawley rats (150-200g, Orient Bio, Gapyeong, Korea) were purchased and bred in SPF (Specific-Pathogen-Free) laboratory which maintains constant temperature (21℃±1℃), humidity (55%) and 12 hours light:dark cycle (light: 07:30~20:00, dark: 20:00~07:30). The rats were supplied with normal feed (Purina Rodent Chow, Purina Co., Seoul, Korea) and water. „

   This paragraph is in 2.3. Surgical procedures. Better to be moved to 2.1. Animals

  1. Figure 5, 6 legend     NG: New Born   ???

6. HCL   ? hydrochloric acid = HCl

Reviewer 3 Report

1.Why authors did not cite work with very similar topics in Tissue Engineering and Regenerative Medicine, Vol. 9, No. 5, pp 276-281 (2012) "Histomorphometric Study on Healing of Critical Sized Defect in Rat Calvaria using Three Different Bovine Grafts" where was the same materials tested by the same method in the same animals and defects? I believe that it is not correct and ethical.

2. From web of producer TiOss clearly resulted the presence of OCP in microstructure of material and it is not clear from experimental method whether authors apllied special procedure for OCP coating or it is origin product from producer.

3 No mean about statistical procedure used in the paper is described in any section.

4. From variances in showed figures  resulted relative large errors in average values of analysed parameters and it is incorrect to show the results in decimals if errors are in percents. Please correct this. Similary in discussion :"....containing OCP were compared, Ti-oss® was 3.63% less at 8 weeks and 7.02% less at 12 weeks. This  could be understood as a result of reflecting the new bone formation..." it is incorrect to say that really exist some difference in presented parameter when the large variances in means were identified.

5. In histological images, I recommend  more detailed description of cells in tissues.

6. I do not see any difference between results from this paper and above showed study published in 2012. If were done some changes in composition of TiOss material (e.g. OCP) - materials with and without chang have to be compared because no conclusions about effect of OCP on bone formation can be given. 

Reviewer 4 Report

This study compares bioabsorption of Ti-OSS and Bio-OSS. The reviewer thinks that it has originality to some extent and acceptable after minor revisions.

(1) In Page 2, Line 71, LeGerose is mistake for LeGeros.

(2) How much is porosity of Ti-OSS and Bio-OSS. Porosity also governs bioabsorption rate.

Round 2

Reviewer 2 Report

The authors have substantially revised the manuscript. All my comments have been covered adequately.

Minor point

Paragraph 2.1.  Redundancy could be removed:

"In our animal experiment, twenty-four 6 week-old female Sprague Dawley rats (150-200g, Orient 85 Bio, Gapyeong, Korea) were used, and six rats were applied per group. Sprague-Dawley rats (150-86 200g, Orient Bio, Gapyeong, Korea) were purchased..."

Reviewer 3 Report

Despite of significant improvement of manuscript that I believe novelty of present results is very doubtful. Besides the results in any case did not show active action of OCP on bone forming properties in TiOss. For answer on this question absolutely different sample for comparison have to be used - TiOss without OCP.

I recommend to show microstructure of TiOss observed by SEM.
